# 12-Month Weight Loss and Adherence Predictors in a Real-World UK Tirzepatide-Supported Digital Obesity Service: A Retrospective Cohort Analysis

**DOI:** 10.3390/healthcare14010060

**Published:** 2025-12-26

**Authors:** Louis Talay, Jason Hom, Tamara Scott, Neera Ahuja

**Affiliations:** 1Faculty of Arts and Social Sciences, University of Sydney, Sydney, NSW 2050, Australia; 2Hospital Medicine Division, Department of Medicine, Stanford University, Stanford, CA 94305, USA; jasonhom@stanford.edu (J.H.); tlscott@stanford.edu (T.S.); nkahuja@stanford.edu (N.A.)

**Keywords:** digital weight-loss service, telehealth, multidisciplinary care, real-world obesity program, chronic care, tirzepatide

## Abstract

**Background**: Obesity management is evolving with the integration of dual GIP/GLP-1 receptor agonists (Tirzepatide) into comprehensive Digital Weight-Loss Services (DWLSs). This model leverages virtual, app-based multidisciplinary care (MDT) to deliver continuous, supervised treatment, distinguishing it from traditional, intermittent clinic-based care. While clinical trials demonstrate high efficacy, real-world data are necessary to evaluate long-term adherence and identify predictive markers for patient persistence in these scalable care models. Specifically, there is a knowledge gap regarding the specific behavioral factors that govern 12-month persistence in these comprehensive, medicated DWLS settings. This study retrospectively assessed the 12-month effectiveness and adherence of a Tirzepatide-supported DWLS and identified demographic, clinical, and behavioral predictors of weight loss and program attrition. **Methods**: Data from 19,693 patients enrolled in the Juniper UK DWLS were analyzed. Adherence was defined by a minimum of 10 medication orders and 12-month weight submission. Weight loss in the full cohort was evaluated using the Last Observation Carried Forward (LOCF) method. Binary logistic and multiple linear regression models identified predictors of adherence and weight loss, respectively, using a comprehensive set of demographic, clinical (e.g., BMI, comorbidities), and behavioral variables. **Results**: The 12-month adherence rate was 27%. The adherent sub-cohort (n = 5322) achieved a mean weight loss of 22.60 (±7.46) percent, compared to 13.62 (±10.85) percent in the full cohort (LOCF). This difference in 12-month mean weight loss was statistically significant (*p* < 0.001). Consistent weekly weight tracking and health coach communication were the strongest positive predictors of long-term adherence and weight loss. Conversely, hyper-engagement, specifically intensive tracking frequency and high weight loss velocity in the first month, was a significant inverse predictor of 12-month adherence. Reporting side effects was positively correlated with adherence, suggesting a reporting bias among engaged patients. **Conclusions**: The DWLS model facilitates the maximum therapeutic effectiveness for adherent patients. However, patient persistence remains the primary translational challenge. As consistent weekly engagement (tracking, coaching) is the strongest predictor of success, clinical strategies should prioritize promoting sustainable, moderate behavioral pacing (i.e., emphasizing consistent weekly engagement over intensive daily tracking and rapid early weight loss) to mitigate attrition risk and optimize the public health effectiveness of medicated DWLSs.

## 1. Introduction

Obesity is a complex, chronic disease characterized by excess adiposity that significantly increases the risk of multiple comorbidities, including type 2 diabetes, cardiovascular disease, and certain cancers [1,2]. The complex etiology, encompassing a dynamic interplay of genetic, neurohormonal, and environmental factors, necessitates a comprehensive and sustained management strategy [3]. While cornerstone treatments involve dietary and physical activity interventions, programs relying solely on these modifications often result in challenges maintaining clinically significant weight loss over extended periods [4].

The landscape of obesity pharmacotherapy has been significantly transformed by the emergence of glucagon-like peptide-1 receptor agonists (GLP-1 RAs), such as semaglutide [5]. These medications, validated in landmark randomized controlled trials (RCTs) like the STEP program, exert their effect by binding to the GLP-1 receptor [6]. This action slows gastric emptying, thereby promoting satiety, and acts centrally in the hypothalamus to reduce appetite, leading to substantial weight reduction [6]. Building on this success, a newer class of medications, dual GLP-1 and glucose-dependent insulinotropic polypeptide (GIP) receptor agonists, exemplified by Tirzepatide, has been developed. Tirzepatide’s simultaneous activation of both the GLP-1 and GIP receptors results in a synergistic metabolic response [7]. This potent mechanism has been shown in large RCTs, such as the SURMOUNT program, to deliver superior mean weight loss compared to first-generation GLP-1 RAs, with participants achieving over 20% weight reduction in extended follow-up [7,8]. The high intrinsic effectiveness of this class is particularly relevant in a scalable Digital weight-loss service (DWLS) model, as the platform’s primary goal is to ensure that the medication’s maximal therapeutic potential is reliably translated into real-world outcomes across a large patient population.

The introduction of these highly effective pharmaceuticals has coincided with the rapid expansion of digital health solutions. Medicated DWLSs offer a critical pathway to scale access to obesity care. These platforms break down common geographic and temporal barriers to continuous medical management [9,10], and virtual care environments have been noted to potentially foster greater comfort and openness in patient-provider communication regarding sensitive health topics [11]. Unique features typically include app-based progress tracking, personalized lifestyle support, and asynchronous, on-demand communication with a multidisciplinary care team (MDT) [9,10]. These latter features, with the possible exception of MDT support, predate medicated DWLSs. Programs such as Noom and Oviva are long-standing DWLS providers that have utilized standalone lifestyle interventions to drive weight loss, i.e., without pharmacotherapy [12,13]. These services remain widely utilized and have demonstrated effectiveness in achieving modest weight loss through habit formation.

With the clinical validation of GLP-1 RAs for weight loss, the landscape has bifurcated. One model involves ‘medication-first’ platforms, which function as streamlined telehealth services focusing on prescription and safety follow-ups [14,15]. In contrast, ‘comprehensive’ models such as Ro attempt to replicate the MDT environment of clinical trials by mandating a combination of medication and intensive lifestyle intervention [16]. The Juniper UK model follows this comprehensive structure. While the operational costs of commercial DWLS models are rarely detailed, it is well established that medicated programs incur significantly higher monthly fees than lifestyle-only services [17].

The UK National Institute for Health and Care Excellence (NICE) emphasizes that pharmacotherapy (delivered via DWLSs or in-person services) must be integrated with comprehensive lifestyle interventions and overseen by an MDT to ensure safe and effective patient-centered care [18]. The Juniper UK DWLS is designed to align with these standards by providing MDT guidance and app-based behavioral support, but collecting real-world data from comprehensive DWLS platforms is essential to evaluate the operational effectiveness of such models. Furthermore, the cost of commercial DWLS models, such as the one studied here, often presents a significant financial barrier, potentially leading to the exclusion of patients from lower socioeconomic backgrounds, a known challenge in scaling access to chronic care services [19].

In this context, collecting real-world data from comprehensive DWLS platforms is essential. While Ng et al. [15] reported 16.5% mean 12-month weight loss in a real-world Tirzepatide cohort, that study’s model was limited to pharmacotherapy and did not include integrated, continuous lifestyle coaching or MDT support. Conversely, Johnson et al. [20] found a substantial difference in 11-month mean weight loss between non-engaged (17.0%) and engaged (21.5%) patients in a comprehensive DWLS, highlighting the therapeutic synergy between medication and high-level behavioral adherence. These findings suggest that identifying predictors of engagement and adherence is crucial for maximizing outcomes. The Juniper UK DWLS is a comprehensive service that combines Tirzepatide with MDT guidance and app-based behavioral support.

This study addresses the existing knowledge gap by retrospectively analyzing a large cohort of patients enrolled in the Juniper UK DWLS to determine the clinical effectiveness (weight-loss outcomes) of the comprehensive Tirzepatide-supported program over 12 months. As a retrospective cohort study, this design allows for the identification of associations and predictive factors but not causal inference. Our primary aims are to evaluate the percentage of patients achieving 12-month program adherence and to calculate the mean weight loss among this adherent cohort. Secondary objectives include comparing outcomes (mean weight loss at 3, 6, 9, and 12 months and side effect incidence) between the full cohort and the adherent sub-cohort and identifying the demographic, clinical, and behavioral factors that predict both weight loss and 12-month adherence.

## 2. Materials and Methods

### 2.1. Study Design

This study retrospectively analyzed data from all patients who commenced the Tirzepatide-supported Juniper UK program (London, UK) between January 1 and 11 November 2024. The latter date was selected on the basis of it falling exactly 12 months after the data extraction date (11 November 2025). All study data were retrieved from the Juniper central data repository on Metabase—an open-source business intelligence tool [21]. The study received human ethics approval from the Just Reasonable Independent Research Ethics Committee on 18 August 2025 (IREC015). Investigators followed the Strengthening the Reporting of Observational Studies in Epidemiology Statement (STROBE) guidelines throughout the study [22].

### 2.2. Program Overview

Prospective patients of the Juniper UK DWLS complete a pre-consultation questionnaire that a pharmacist independent prescriber uses to determine their eligibility for the service. These questionnaires contain up to 100 questions and are often supported with medical imaging, pathology results, and/or reports from previous clinicians, should prescribers request them.

Prescribing practitioners base their Juniper UK eligibility decisions on the Mounjaro medication guide, which details body mass index (BMI) ranges, contraindications, and drug interactions. Inclusion criteria include a BMI of at least 27 kg/m^2^ for patients of non-Caucasian ethnicity, a BMI of at least 30 kg/m^2^ for everyone else, plus at least one weight-related comorbidity (e.g., hypertension, obstructive sleep apnea). Hard exclusion criteria included the following contraindications: multiple endocrine neoplasia syndrome type 2, a personal or family history of medullary thyroid cancer, acute kidney disease, acute gallbladder disease, acute pancreatitis, hypoglycemia, severe gastrointestinal disease, and a known hypersensitivity to Tirzepatide or any of the product’s components.

All prescribing actions are automatically recorded in the Juniper clinical auditing repository on Jira. This system utilizes data analytics to flag potential errors or safety issues for auditors. Patients enrolled in the Juniper UK DWLS are assigned an MDT, which includes a pharmacist independent prescriber, a university-qualified health coach, a pharmacist, and a medical support officer. Interactions between the patient and their MDT occur via the Juniper mobile application. All patient-MDT communications and program entries (such as weight and progress tracking) are automatically uploaded to the central data repository for efficient care coordination and data management. To enhance data accuracy, the Juniper app employs input validation, flagging implausible weight entries for manual review by the MDT. Additionally, prescribing clinicians mandate weight submission at the 5-month consultation, providing a clinical verification point for patient-reported data.

During the study period, the Juniper UK DWLS only provided medication-supported therapy, i.e., it has never offered standalone lifestyle or GLP-1 RA/dual GIP/GLP-1 RA treatment. After paying the first monthly subscription fee, Juniper patients receive their initial one-month supply of medication along with access to personalized lifestyle coaching. This coaching is delivered through the Juniper app and provides multimodal educational content, progress trackers, and customized meal and exercise plans. Patients continue to receive monthly medication until (and including) month 4, unless they discontinue payment or choose to opt out of the program. Prescribing clinicians are mandated to assess patients at the 5-month mark of their treatment journey. Patients must be approved by their prescribing clinician at this 5-month follow-up consultation to receive their 5th medication order and continue with lifestyle coaching. Although the MDT encourages patients to log weight data fortnightly via the app, this follow-up consultation is the first time since the program started that weight data is a mandatory requirement for continued participation. Patients are instructed to notify their MDT immediately if any side effects occur.

### 2.3. Medication Titration Schedule

Juniper’s prescribing clinicians adhere to the Mounjaro medication guidelines to establish patient titration schedules. The standard recommended schedule is: 2.5 mg once weekly for four weeks, followed by increases of 2.5 mg every four weeks, up to a maximum of 15 mg if required (for example, 5 mg once weekly for weeks 5–8, 7.5 mg once weekly for weeks 9–12, and so on). However, prescribers retain the clinical judgment to postpone up-titration if a patient’s condition makes the standard schedule inappropriate. This clinical judgment involves a risk-benefit assessment: While the standard schedule aims for timely therapeutic effect, prescribers prioritize patient safety and long-term program adherence. A deviation, such as postponing up-titration due to adverse events or missed doses, serves to mitigate the risk of severe or persistent side effects that could otherwise lead to treatment intolerance and discontinuation, thereby undermining therapeutic success.

### 2.4. Program Cost

Monthly fees for the Juniper UK DWLS varied by the patient’s medication dose, ranging from 149 to 339 Great British Pounds. Due to the lack of income data in this retrospective analysis, we were unable to assess whether the higher cost associated with higher Tirzepatide doses acted as a financial barrier contributing to socioeconomic differences between the adherent and non-adherent cohorts.

### 2.5. Endpoints

The study’s primary endpoints were the percentage of patients who reasonably adhered to the Juniper program over a 12-month period and the mean weight loss among these patients. Adherence was determined by the following criteria: patients received a minimum of 10 Tirzepatide orders and submitted weight data between 355 and 375 days after program initiation (10 days either side of 12 months (365 days)). The ≥10 order threshold was selected to ensure adequate treatment exposure while accounting for patients who may have paused or delayed treatment for a short period. It is important to note that, given the asynchronous nature of the DWLS, medication adherence is measured via the count of pharmacy orders, which serves as a proxy for medication consumption. Actual self-administration (injection) is not directly monitored.

Secondary endpoints included a comparison of side effect incidence and the mean weight loss at 3, 6, 9 and 12 months between the full and adherent cohorts. All missing weight data in the full cohort were imputed via the last observation carried forward (LOCF) method. It is acknowledged that the LOCF method introduces a known conservative bias, likely underestimating the true mean weight loss for patients who discontinue early. Specifically, this approach fails to capture the common physiological phenomenon of weight rebound that often occurs after the cessation of medicated weight-loss therapy, thereby leading to a more favorable (but potentially inaccurate) mean outcome in the full cohort. The analysis also sought to understand the factors influencing weight loss in the adherent cohort and 12-month adherence in the full cohort. These factors ranged from demographic variables such as initial BMI and comorbidities to behavioral variables such as weight tracking frequency, health coach message frequency, and maximum Tirzepatide dosage. Weight tracking frequency and health coach communication frequency were calculated by dividing the number of weeks in which a patient tracked or communicated at least once by the number of weeks they were on the program, multiplied by 100 (e.g., a patient was on the program for 8 weeks, and tracked their weight 10 times across weeks 4 and 5 week only: 2/8 × 100 = 25%).

### 2.6. Statistical Analysis

Means and standard deviations reported for descriptive data. Multiple linear regression was performed to assess the effect of demographic, clinical and behavioral variables on 12-month weight loss. Continuous variables (age, initial BMI, initial weight) were compared using the independent-samples Welch’s *t*-test and categorical variables (sex at birth, ethnicity, and comorbidities) were compared using the Chi-squared (x^2^) test to determine if baseline distributions differed significantly between the adherent and full cohorts. Binary logistic regression was run to identify the predictors of 12-month adherence (yes/no). For both models, certain numeric variables such as initial BMI, month-1 track count and initial comorbidities were dichotomized or categorized into factor variables wherever investigators determined this would add important clinical nuance to the results. Side effect incidence (yes/no) was compared between the full and adherent cohorts using an Exact Binomial test. All data were extracted from the Juniper central data repository on Metabase using SQL. Data were then uploaded to RStudio, version 2023.06.1 + 524 (RStudio: Integrated Development Environment for R, Boston, MA, USA) for all analyses and visualizations. Model assumptions, including linearity, independence, and homoscedasticity for the linear model and the link function for the logistic model, were assessed and met. No formal sensitivity analyses were conducted, given the retrospective design and the large sample size.

## 3. Results

### 3.1. Descriptive Data

In total, 19,693 patients commenced the Tirzepatide-supported Juniper UK program between 1 January and 11 November 2024. Among these patients, 5322 (27%) satisfied the 12-month program adherence criteria. A further 2688 (13.65%) were relatively adherent to their medication schedule, receiving a minimum of 10 orders of Tirzepatide, but did not submit weight data within the 355–375-day window. Of the remaining 11,683 (59.32%) patients, 1131 (9.68%) achieved a healthy BMI or their target weight. The full patient flow is visualized in Figure 1. Across the full cohort, mean patient age was 43.10 (±11.9) years and mean initial BMI was 34.96 (±6.91) kg/m^2^ (Table 1). Nearly 90 percent (89.69) of patients were female at birth and 16,150 (82%) were of Caucasian ethnicity. Statistically significant baseline differences were observed between the adherent and full cohorts. The adherent cohort was older and had a higher proportion of male patients and a higher proportion of Caucasian patients, suggesting a selection effect for long-term persistence in the program.

In the adherent cohort, the mean 12-month weight loss was 22.60 (±7.46) percent, with 5261 (98.85%) patients losing a minimum of 5 percent of their baseline weight and 5084 (95.53%) reaching the 10 percent milestone (Table 2). In the full cohort, which used the LOCF method to impute missing data, the mean 12-month weight loss figure was 13.62 (±10.85) percent. Similar differences were observed across the other 3 intervals: 3-month, 6-month and 9-month weight loss (Figure 2). Regarding side effects, 16,356 (83.05%) patients in the full cohort reported at least one adverse event, versus 5084 (95.53%) patients in the adherent cohort. A one-sample Exact Binomial Test revealed that this difference was statistically significant (CI = [0.949, 0.961], *p* < 0.001). Among the patients who reported side effects, the distribution of worst side effect severity was comparable across the two cohorts, with mild being the most common level in both (~57–58%).

### 3.2. Weight-Loss Covariates

A generalized linear regression model was run to predict 12-month weight loss in the adherent cohort, explaining 39.94% of the variance in the outcome (Appendix A). Statistically significant predictors were found across demographic, clinical, and behavioral factors. Regarding demographics, age was negatively associated with 12-month weight loss (*β* = −0.10, *p* < 0.001), and being male at birth was also associated with a significant decrease (*β* = −3.54, *p* < 0.001). The ethnicity binary also had a weak but statistically significant association with weight-loss, with white patients tending to lose more weight than non-white patients (*β* = 0.62, *p* < 0.05). Relative to the most populated BMI group (30–34.99 kg/m^2^), the lowest category (<30 kg/m^2^) lost significantly less weight (*β* = −4.16, *p* < −0.001), whereas the three higher BMI categories (35–39.99; 40–44.99; >45 kg/m^2^) lost statistically more weight (*β* = 2.34, *p* < −0.001; *β* = 2.36, *p* < −0.001; *β* = 1.04, *p* < −0.001).

Behavioral engagement was also found to be a key factor, as a higher weekly weight track percentage was positively associated with 12-month weight loss (*β*: 0.084, *p* < 0.001). Conversely, weight tracking in the first month was negatively correlated with weight loss in the three highest categories, relative to patients who only tracked once (baseline weight). Patients in the 16–25 weight entry and >25 weight entry categories lost an average over 4 percentage points less than the baseline group (both *p* < 0.001), while the reduction in the 11–15 weight entries category was over 3 percentage points (*p* < 0.01). Early progress was strongly predictive, with one-month weight-loss percentage showing a strong positive association (*β* = 0.93, *p* < 0.001) with 12-month weight loss. Similarly, achieving clinically significant weight loss at month 3 (≥5%) correlated with a substantial increase in the outcome (*β*: 5.00, *p* < 0.001), as did the attainment of a healthy BMI (<25 kg/m^2^) or their goal weight (*β*: 4.18, *p* < 0.001).

Relative to the highest maximum Tirzepatide dose (15 mg), significant differences were observed in the 7.5 mg (*β* = −1.30, *p* < 0.001), 5 mg (*β* = −2.49, *p* < 0.001) and 2.5 mg (*β* = −2.95, *p* < 0.05) categories but not in the 10 mg and 12.5 mg categories. The presence of baseline comorbidities, side effect incidence and weekly health coach messaged percentage were not found to be statistically significant predictors of 12-month weight-loss.

### 3.3. Program Adherence Covariates

A generalized linear model (Logistic Regression, binomial family) was run to predict 12-month adherence (Adherent vs. Churned), explaining 23.91% of the variability in the outcome R^2^ = 0.2391). Statistically significant predictors were found across demographic, clinical, and behavioral factors (Appendix B). Older age (*β* = 0.01, *p* < 0.01), being male at birth (*β* = 0.38, *p* < 0.001), and White ethnicity (*β* = 0.13, *p* < 0.05) were all associated with a statistically significant increase in the log-odds of 12-month adherence. Regarding BMI categories (relative to the reference group of 30–34.99 kg/m^2^), only the highest BMI group (>45 kg/m^2^) was significantly associated with a decrease in the log-odds of adherence (*β* = −0.27, *p* < 0.001). Comorbidities showed marginal negative associations, with patients reporting three or more comorbidities having a *β* = −0.20 (*p* = 0.056) and those reporting two comorbidities having a *β* = −0.13 (*p* = 0.059).

Early success markers showed mixed effects, highlighting a distinction between velocity and sustainability. Achieving significant weight loss at month 3 (≥5%) was associated with an increase in the log-odds of adherence, though the relationship was marginally significant (*β* = 0.15, *p* = 0.06). Conversely, 1-month weight loss velocity was negatively associated with adherence in the highest loss categories, relative to the reference group (2.51–5% loss). Patients with >10% loss at 1 month had a *β* = −0.49 (*p* < 0.001), meaning they were 1.67 times more likely to churn, and patients with 7.51–10% loss were 1.36 times more likely to churn.

Behavioral engagement showed the strongest correlations: weekly weight track percentage was the most powerful predictor, with a large positive association with adherence (*β* = 0.048, *p* < 0.001). However, high tracking frequency in the first month was strongly and negatively associated with adherence, consistent with the weight loss velocity finding. Patients in the >25 weight entry category had the largest negative coefficient (*β* = −1.73, *p* < 0.001), followed by the 16–25 entry category (*β* = −1.32, *p* < 0.001), the 11–15 category (*β* = −1.06, *p* < 0.001) and the 6–10 category (*β* = −0.98, *p* < 0.01) (Figure 3). The model demonstrated a strong positive association between weight outcome attainment and perseverance: attaining a healthy BMI or the patient’s target weight was highly predictive of 12-month adherence (*β* = 0.48, *p* < 0.001). Patients who attained this outcome were 1.62 times more likely to be adherent at 12 months than those who did not. Reporting any side effects was also unexpectedly and significantly associated with an increase in the log-odds of adherence (*β* = 0.47, *p* < 0.001). Finally, weekly health coach messaged percentage was a strong positive predictor of adherence (*β* = 0.027, *p* < 0.001), whereas previous use of modern weight-loss medications (Semaglutide, Liraglutide or Tirzepatide) was a negative predictor (*β* = −0.35, *p* < 0.001).

## 4. Discussion

This retrospective cohort study of 19,693 patients enrolled in the Tirzepatide-supported Juniper UK DWLS provides essential real-world evidence regarding 12-month effectiveness and adherence predictors in a comprehensive digital obesity setting.

A primary finding is the mean 12-month weight loss of 22.60 (±7.46) percent observed in the adherent cohort (n = 5322). This outcome demonstrates an effectiveness level comparable to the mean weight loss achieved in highly controlled RCTs of Tirzepatide [8], and the 11-month figure reported in the engaged cohort of the Johnson et al. study (M = 21.5%). The results align with NICE guidelines on combining dual GIP/GLP-1 RA pharmacotherapy with comprehensive, app-based MDT support. Furthermore, the finding that 98.85 percent of adherent patients achieved the clinically significant 5 percent weight-loss threshold suggests that medicated DWLSs deliver reliable outcomes to adherent patients. Furthermore, the finding that 98.85 percent of adherent patients achieved the clinically significant 5 percent weight-loss threshold suggests that medicated DWLSs deliver reliable outcomes to adherent patients. This aligns with published narrative syntheses describing Tirzepatide’s dual GIP/GLP-1 receptor mechanism as a key driver of its robust weight-loss effects across diverse clinical trials and obesity-management settings [23].

However, the contrast between the adherent cohort’s success and the full cohort’s LOCF mean weight loss of 13.62 (±10.85) percent highlights the major public health challenge of adherence. With only 27 percent of patients satisfying the strict 12-month adherence criteria, the significant difference in outcomes reveals that while the medication is highly effective, the benefits are limited to those who successfully persist through the duration of the program. Transparent reporting of both the adherent and full (LOCF) cohorts is thus essential for accurately conveying a program’s overall impact.

The predictive models revealed that behavioral engagement is the most powerful determinant of long-term success. The positive association between weekly weight track percentage and both 12-month weight loss (*β* = 0.084, *p* < 0.001) and adherence (*β* = 0.048, *p* < 0.001) confirms the findings of Johnson et al. [20] and broader DWLS literature that sustained, consistent engagement with tracking tools is crucial for facilitating self-monitoring and reinforcing habit change [18]. Similarly, the strong positive relationship between weekly health coach message percentage and adherence (*β* = 0.027, *p* < 0.001) demonstrates that active, ongoing MDT communication is a key component in preventing patient dropout, possibly by resolving issues quickly or providing motivation.

A novel and critical finding is the negative association between hyper-engagement during the first month (specifically high weight tracking frequency and high initial weight loss velocity) and 12-month adherence. Patients in the highest weight entry categories were substantially more likely to churn, and patients achieving >10% weight loss at month 1 were 1.67 times more likely to churn than the reference group. Importantly, this trend was observed even while controlling for the attainment of a health BMI (<25%) or a patient’s target weight, which was positively correlated with 12-month adherence. These discoveries suggest that while early success is a motivator, excessive early weight-loss may lead to unsustainable habits or rapid goal attainment followed by subsequent disengagement—a phenomenon often termed “burnout.” The findings specific to first month weight tracking were even starker. Not only did patients in the highest three tracking categories (>25; 16–25; 11–15) experience significantly lower weight loss than the baseline group (1 weight entry), but they were also significantly less likely to adhere to the program for 12 months (along with patients in the 6–10 category). While this discovery lacks clear precedent in the wider chronic disease management literature, it may stem from a combination of heightened performance anxiety and misaligned expectations concerning the speed of weight-loss outcomes [24,25]. These expectations have possibly been exaggerated by success narratives surrounding new weight-loss pharmaceuticals on mainstream and social media [26,27]. When this intense initial commitment does not immediately translate into continued exponential loss, the resulting tracking fatigue or disillusionment can lead to disengagement and program dropout. It must be noted that these proposed psychological mechanisms (burnout, performance anxiety) are hypothesized based on the observed statistical associations and prior literature and were not empirically tested or directly measured in this retrospective dataset. Nevertheless, DWLS clinicians may view these findings on hyper initial engagement and adherence as an opportunity to coach patients toward sustainable, moderate pacing. Specific interventions could include setting realistic, long-term weight-loss goals, emphasizing consistent moderate weekly tracking over intensive daily tracking, and developing automated alerts to flag patients who exhibit high initial weight-loss velocity, allowing for timely, supportive coach intervention to manage expectations and mitigate burnout.

The counter-intuitive finding that reporting any side effects was significantly associated with an increase in the log-odds of 12-month adherence (*β* = 0.47, *p* < 0.001) must be interpreted within the comprehensive care model. Adherent patients, who are consistently engaged with the program, are the most likely to diligently report adverse events to their MDT via the app, as instructed. Patients who churned may have stopped using the app due to side effects without formally recording them. Therefore, this correlation likely reflects an adherence bias in reporting, rather than a protective effect of side effects on perseverance. Clinically, this result emphasizes the importance of a structured reporting pathway (like the app) for capturing safety data among engaged users. It also highlights the utility of exit surveys as a means of monitoring the proportion of patients whose side effects were the primary factor behind their discontinuation. Unfortunately, data from Juniper exit surveys were too incomplete to incorporate into this analysis—a limitation affected by the legal restriction on mandating their completion. Future program iterations should explore incentivizing the completion of exit surveys (e.g., small rewards) or utilizing brief, structured, automated messaging sequences (e.g., SMS/email) triggered immediately upon app cessation, which may yield higher response rates for capturing discontinuation reasons. Alternatively, prospective studies could be run, whereby investigators call every patient who stops the program early to record their primary discontinuation reason.

Regarding demographic factors, older age (*β* = −0.10, *p* < 0.001) and male sex at birth (*β* = −3.54, *p* < 0.001) were negative predictors of weight loss, aligning with commonly observed trends in weight management literature. Older age is known to present unique barriers to sustained weight loss, as research suggests that increased work and family commitments can hinder the maintenance of healthy diets [28,29], while physiological changes such as pre-existing sarcopenia and a lower basal metabolic rate may intrinsically limit the potential for weight reduction in older populations [30,31].

Additionally, the greater percentage weight loss observed in patients with higher initial BMI (relative to the 30–34.99 kg/m^2^ group is expected) likely reflects the ceiling effect of percentage weight loss in individuals with greater initial weight [32]. Interestingly, the opposite trend has been observed in a previous study of the Juniper program, although this may have been due to its shorter duration (16 weeks) or the fact that the association was not controlled both other key covariates [33]. Regarding the medication titration, the finding that intermediate doses (10 mg and 12.5 mg) did not significantly differ from the highest dose (15 mg) in predicting 12-month weight loss warrants further investigation. This suggests that the maximum weight loss response may plateau at or before the 10 mg dose, or alternatively, that the statistical power within these specific dose strata was insufficient to detect a smaller difference when compared to the reference group. The negative association between previous use of modern weight-loss medications (GLP-1 RA or dual agonist) and 12-month adherence (beta = −0.35, *p* < 0.001) is notable. This correlation may indicate that this sub-population, being less treatment-naïve, may harbor higher, potentially unrealistic expectations about the speed or extent of weight loss, or they may represent patients who are prone to cycling through various pharmacological treatments without establishing long-term behavioral persistence. Finally, the discovery that a disproportionate number of non-Caucasian patients failed to adhere to the program for 12 months is consistent with previous research on DWLSs and chronic care services in general [33,34]. Although income-related data were not available, It is likely that this trend is a reflection of Juniper UK’s high monthly fee. Beyond cost, potential structural factors contributing to lower adherence may include differences in digital health literacy, access to high-speed internet, or a lack of culturally tailored MDT support and resources [31]. Addressing these disparities may require structural interventions, such as exploring financial support or tiered pricing models for underserved populations.

This study’s interpretation is limited by its retrospective design, which can only establish correlations, not causation and is subject to the inherent risk of unmeasured confounding from unrecorded variables (e.g., patient-reported dietary data, true socioeconomic status) that may influence both adherence and weight loss. Although the scale of the cohort is a major strength, the reliance on LOCF for the full cohort introduces a known conservative bias that may not accurately reflect the mean weight loss achieved by those who dropped out early. The sample was also overrepresented by female and Caucasian patients, and given the program fee, would have likely excluded patients of lower socioeconomic status. These factors limit the study’s generalizability. Future studies should investigate the role of culturally tailored support and financial barriers in DWLS adherence among diverse populations. Furthermore, the analysis relies on patient-reported data, such as weight entries and side effect reports, which may be subject to desirability or measurement bias, and medication adherence was measured via the count of pharmacy orders, which was only a proxy for medication consumption and therefore may not have been completely accurate. Future prospective studies should employ more robust longitudinal techniques, such as multiple imputation, to more accurately handle missing data. Finally, analyses would have been enriched by discontinuation reason data, which investigators did not have access to.

## 5. Conclusions

This retrospective analysis suggests that the Tirzepatide-supported DWLSs are capable of delivering good long-term weight-loss outcomes, with the adherent cohort achieving a mean 12-month loss of 22.60 (±7.46) percent. This finding validates NICE guidelines of using pharmacotherapy as a supplement to continuous multidisciplinary care.

However, the primary translational challenge lies in patient retention, as evidenced by the low (27%} adherence rate and the subsequent reduction in mean weight loss to 13.62% across the full intent-to-treat cohort. These data underscore that the substantial benefits of Tirzepatide are contingent upon persistence in the program, making adherence the single most critical public health hurdle for digital obesity services. Furthermore, the findings underscore the need for future program development to include equity-focused interventions to address socioeconomic and demographic barriers to long-term persistence.

The study’s predictive models offer preliminary insights that can help inform clinical decision-making for overcoming this challenge. The strongest predictors of success were not initial results but markers of long-term consistency: high weekly weight-tracking percentage and frequent health coach communication. Crucially, the discovery that hyper-engagement and high-velocity weight loss in the first month are negatively associated with 12-month adherence suggests that clinicians should actively intervene to manage performance anxiety and temper potentially unrealistic patient expectations fueled by public discourse. Coaching strategies should focus on sustainable, moderate pacing rather than intensive, rapid initial weight loss to mitigate the risk of burnout and subsequent dropout.

In conclusion, while medicated DWLSs deliver reliable outcomes to adherent patients, their overall effectiveness relies on improving long-term adherence. Future prospective studies should investigate targeted interventions designed to promote consistent, moderate engagement throughout the initial months of treatment to maximize patient persistence and optimize therapeutic outcomes at scale.

Future prospective studies are needed to establish causation and test targeted adherence interventions. Methodologically, these studies should employ alternative imputation techniques, such as multiple imputation, to more accurately handle missing data. Furthermore, future work must prioritize recruiting a more diverse and representative sample and should incorporate robust mechanisms to prospectively capture patient discontinuation reasons and detailed socioeconomic data.

## Figures and Tables

**Figure 1 healthcare-14-00060-f001:**
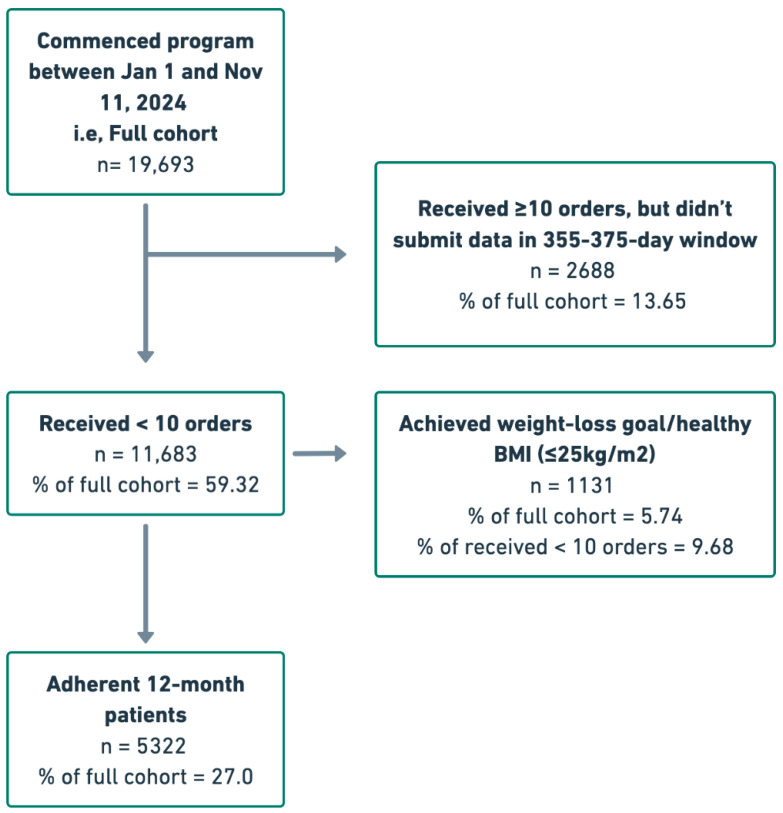
Patient flow chart.

**Figure 2 healthcare-14-00060-f002:**
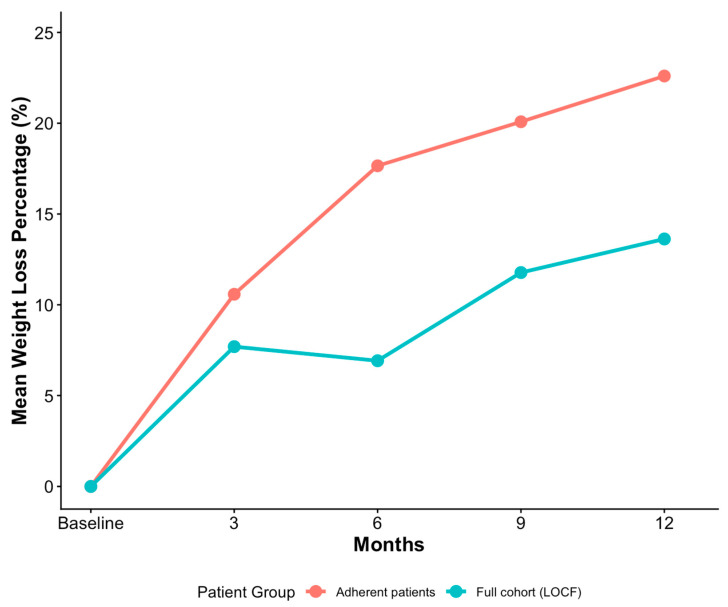
Monthly weight-loss in adherent patients compared to full cohort (LOCF).

**Figure 3 healthcare-14-00060-f003:**
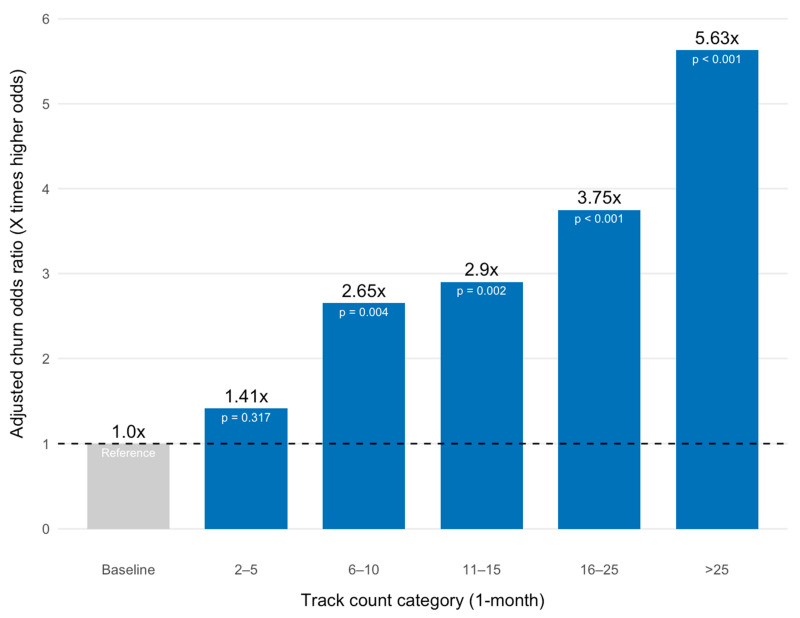
Model-Adjusted Churn Odds Ratio by Month-1 Tracking Frequency.

**Table 1 healthcare-14-00060-t001:** Patient demographic data.

Demographic Variable	Full 12-Month Cohort	Adherent 12-Month Cohort	*p*-Value
Total patients	19693	5322	
Age, Mean (SD) years	43.10 (±11.90)	45.59 (±10.90)	<0.001
**Sex at birth, n (%)**			
Female	17,663 (89.69)	4910 (92.26)	<0.001
Male	2030 (10.31)	412 (7.74)	
**Ethnicity, n (%)**			
Caucasian	16,150 (82.0)	4641 (87.20)	<0.001
Asian (including subcontinent)	1716 (8.71)	385 (7.23)	<0.01
Mixed ethnicity	974 (4.95)	133 (2.50)	<0.01
African/Caribbean	401 (2.04)	103 (1.94)	0.72
Middle Eastern	207 (1.05)	21 (0.39)	<0.05
Latino/Hispanic	127 (0.64)	28 (0.51)	0.38
Pacific Islander/Māori	118 (0.60)	11 (0.21)	<0.05
**Clinical information, Mean (SD)**			
Initial BMI—kg/m^2^	34.96 (±6.90)	35.71 (±7.10)	<0.001
Initial Weight—kg	96.72 (±21.15)	98.90 (±21.39)	<0.001

**Table 2 healthcare-14-00060-t002:** Weight-loss and side effect outcomes.

	Adherent 12-Month Cohort	Full 12-Month Cohort
**Patient Count**	5322	19,693
**Mean 3-month weight loss, % (SD)**	10.58 (±4.07)	7.69 (±5.91)
**Mean 6-month weight loss, % (SD)**	17.65 (±5.5)	6.92 (±7.21)
**Mean 9-month weight loss, % (SD)**	20.08 (±5.75)	11.77 (±9.63)
**Mean 12-month weight loss, % (SD)**	22.60 (±7.46)	13.62 (±10.85)
**12-month weight-loss milestones**		
Lost ≥ 5%—n (%)	5261 (98.85)	13762 (69.88)
Lost ≥ 10%—n (%)	5084 (95.53%)	11,844 (60.14)
**Reported Side Effects (%)**	5084 (95.53%)	16,356 (83.05)
**Worst side effect severity (% total side effects)**		
Mild	2965 (58.32)	9427 (57.64)
Moderate	1712 (33.68)	5899 (36.08)
Severe	407 (8.00)	1030 (6.30)

## Data Availability

The raw study data will be made available by authors upon reasonable request. The data are not publicly available due to privacy and ethical restrictions.

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
