# Peer review of "12-Month Weight Loss and Adherence Predictors in a Real-World UK Tirzepatide-Supported Digital Obesity Service: A Retrospective Cohort Analysis"

_healthcare, 2025, doi:10.3390/healthcare14010060_

Round 1
Reviewer 1 Report
Comments and Suggestions for Authors
The study builds on prior research, such as the Johnson et al. and Ng et al. studies, by providing a more comprehensive analysis of predictors of adherence and weight loss in a real-world DWLS setting. It also adds new dimensions to the understanding of early engagement and its impact on long-term outcomes, which has not been extensively explored in previous studies. The manuscript is well written; however, I made numerous suggestions to improve its quality.
Title
- The title could be more specific about the study's focus on predictors of adherence and weight loss. ​ For example: "Predictors of 12-Month Weight Loss and Adherence in a Real-World Tirzepatide-Supported Digital Obesity Service." ​
- Including "Juniper UK DWLS" in the title could make it more specific and distinguish the study from other DWLS research.
Abstract
- The abstract could briefly mention the key finding that hyper-engagement and high initial weight-loss velocity negatively impact long-term adherence, as this is a novel and critical insight from the study. ​
- The abstract could also emphasize the importance of behavioral engagement (e.g., weekly weight tracking and health coach communication) as the strongest predictors of success, a central theme of the manuscript.
Introduction
Please consider adding the following to improve.
- Expand on DWLS Models: While the manuscript emphasizes integrating pharmacotherapy with lifestyle interventions and multidisciplinary care, it could provide more detail on how DWLS platforms differ from traditional weight-loss programs. For example, the authors could elaborate on the unique features of DWLS, including app-based tracking, communication with health coaches, and personalized lifestyle plans.
- Comparison with Other DWLS: The background could benefit from a more detailed comparison of the Juniper UK DWLS with other DWLS models (if any), particularly regarding their structures, costs, and integration of pharmacotherapy and lifestyle interventions.
- Address Socioeconomic Factors: The manuscript briefly notes the potential exclusion of patients from lower socioeconomic backgrounds due to the program's high cost. ​ Expanding on how socioeconomic factors impact adherence and outcomes could provide a more holistic understanding of the challenges in scaling DWLS.
- Behavioral Factors: The background could introduce the concept of behavioral engagement (e.g., weight tracking and health coach communication) as a key factor in DWLS success, given its significant focus in the study.
Methods
The following area needs improvement in the method section.
- Add references: Please add suitable references for Juniper Central Data Repository in Metabase and the STROBE guidelines.
- Data Collection Tools: While the manuscript mentions the use of the Juniper app and Metabase for data collection, it could provide more detail on how these tools were validated and whether measures were taken to ensure data accuracy (e.g., verifying patient-reported weight entries).
- Medication Titration Schedule: The titration schedule is described, but the rationale for deviations from the standard schedule (e.g., postponing up-titration due to adverse events) could be further clarified to explain how these decisions were made. ​
- Handling Missing Data: The use of the Last Observation Carried Forward (LOCF) method for imputing missing weight data is mentioned, but the limitations of this approach (e.g., potential bias) could be discussed more thoroughly in the methods section rather than in the discussion. ​
- Behavioral Metrics: The calculation of behavioral engagement metrics (e.g., weekly weight tracking percentage and health coach message frequency) is described, but the exact process for extracting and analyzing these data from the app is not detailed. ​ This could be clarified further.
- Statistical Analysis: While the statistical methods are described, the manuscript could provide more detail on how the models were validated, including whether any sensitivity analyses were conducted to assess the robustness of the findings.
Results
Table 2 lists "Mild" side effects as the most common level (~57-58%), but the percentages for "Moderate" and "Severe" side effects are missing. ​ This is an omission and should be corrected for completeness.
Figure 1: The percentages in the flow chart are inconsistent with the written text.
- The relatively adherent cohort, the percentage is listed as 13.65% of the root, which is unclear. ​ The term "root" is not defined in the document, and it may confuse readers. It would be better to clarify this as "13.65% of the full cohort." ​
- Similarly, for the group that achieved a healthy BMI or target weight, the percentage is listed as 9.68% of the root. ​ This should also be clarified as "9.68% of the full cohort."
Figure 2 shows the weight-loss trends for the adherent and full cohorts, but the line labels could be more transparent. ​
- The labels "Adherent patients" and "Full cohort" are correct. Still, it would be helpful to specify that the full cohort data are based on the LOCF (Last Observation Carried Forward) method, as this is mentioned in the text but not in the figure.
Figure 3 shows the churn odds ratio for different tracking categories, but the baseline group (1 weight entry) is labeled "1x" with a p-value of 1, which is incorrect. ​ A p-value of 1 is not meaningful in statistical analysis. This may be a typographical error and should be corrected.
Discussion
The discussion is well written and aligns with the study's findings and objectives, but it could benefit from more actionable recommendations as follows:
- Specific strategies to improve adherence include: designing interventions to address "burnout" caused by hyper-engagement in the first month, exploring ways to manage patient expectations about weight-loss velocity, and offering financial support or tiered pricing to address socioeconomic barriers for non-Caucasian patients.
- More context on how DWLS clinicians can address this issue, such as educating patients about realistic weight-loss trajectories, implementing pacing strategies to encourage moderate engagement and sustainable habits, and developing tools to identify and support patients at risk of burnout early in the program.
- Potential interventions to address these disparities include offering culturally tailored support and resources, investigating socioeconomic factors more thoroughly in future studies, and exploring partnerships with public health organizations to subsidize costs for underserved populations.
- Propose specific improvements for future studies, such as conducting prospective studies to establish causation and using alternative methods to handle missing data, such as multiple imputation. Recruiting a more diverse and representative sample to improve generalizability. Collecting more complete data on discontinuation reasons and socioeconomic factors
- Evidence-based strategies for DWLS providers include setting realistic weight-loss goals for patients, monitoring early engagement, and intervening when hyper-engagement is detected. ​Enhancing health coach training to address performance anxiety and burnout, and developing targeted interventions for patients with high initial BMI or multiple comorbidities.
Author Response
The study builds on prior research, such as the Johnson et al. and Ng et al. studies, by providing a more comprehensive analysis of predictors of adherence and weight loss in a real-world DWLS setting. It also adds new dimensions to the understanding of early engagement and its impact on long-term outcomes, which has not been extensively explored in previous studies. The manuscript is well written; however, I made numerous suggestions to improve its quality.
Comment 1:
Title
- The title could be more specific about the study's focus on predictors of adherence and weight loss. ​ For example: "Predictors of 12-Month Weight Loss and Adherence in a Real-World Tirzepatide-Supported Digital Obesity Service." ​
- Including "Juniper UK DWLS" in the title could make it more specific and distinguish the study from other DWLS research.
Response 1: Thank you for this excellent recommendation. We have now incorporated elements of your suggestion with that of another reviewer to develop the following title: 12-Month Weight Loss and Adherence Predictors in a real-world UK Tirzepatide-Supported Digital Obesity Service: A Retrospective Cohort Analysis
Comment 2:
Abstract
- The abstract could briefly mention the key finding that hyper-engagement and high initial weight-loss velocity negatively impact long-term adherence, as this is a novel and critical insight from the study. ​
- The abstract could also emphasize the importance of behavioral engagement (e.g., weekly weight tracking and health coach communication) as the strongest predictors of success, a central theme of the manuscript.
Response 2: Thank you for this insightful comment. We have now changed sentences 2 and 3 of the results section of the abstract to the following:
“Consistent weekly weight tracking and health coach communication were the strongest positive predictors of long-term adherence and weight loss. Conversely, hyper-engagement, specifically intensive tracking frequency and high weight loss velocity in the first month, were significant inverse predictors of 12-month adherence.”
We also added this clause to the conclusion section of the abstract:
“As consistent weekly engagement (tracking, coaching) is the strongest predictor of success, clinical…”
Comment 3:
Introduction
Please consider adding the following to improve.
- Expand on DWLS Models: While the manuscript emphasizes integrating pharmacotherapy with lifestyle interventions and multidisciplinary care, it could provide more detail on how DWLS platforms differ from traditional weight-loss programs. For example, the authors could elaborate on the unique features of DWLS, including app-based tracking, communication with health coaches, and personalized lifestyle plans.
Response 3:
Thank you for this important comment. We have now added the following sentence to the third paragraph of the introduction:
“Unique features typically include app-based progress tracking, personalized lifestyle support, and asynchronous, on-demand communication with a multidisciplinary care team (MDT) [9,10].”
Comment 4:
- Comparison with Other DWLS: The background could benefit from a more detailed comparison of the Juniper UK DWLS with other DWLS models (if any), particularly regarding their structures, costs, and integration of pharmacotherapy and lifestyle interventions.
Response 4: Thank you for this comment.
Comment 5:
- Address Socioeconomic Factors: The manuscript briefly notes the potential exclusion of patients from lower socioeconomic backgrounds due to the program's high cost. ​ Expanding on how socioeconomic factors impact adherence and outcomes could provide a more holistic understanding of the challenges in scaling DWLS.
Response 5: Thank you for this important comment. We have now added the following sentence to the end of the 3rd paragraph of the introduction:
“Furthermore, the cost of commercial DWLS models, such as the one studied here, often presents a significant financial barrier, potentially leading to the exclusion of patients from lower socioeconomic backgrounds, a known challenge in scaling access to chronic care services [13]”
Comment 6:
- Behavioral Factors: The background could introduce the concept of behavioral engagement (e.g., weight tracking and health coach communication) as a key factor in DWLS success, given its significant focus in the study.
Response 6:
Thank you for this suggestion. We agree that establishing the importance of behavioral engagement is essential. We have now added the following 11 lines (in bold) to the third paragraph of the introduction to add further context around behaviorual engagement:
“Unique features typically include app-based progress tracking, personalized lifestyle support, and asynchronous, on-demand communication with a multidisciplinary care team (MDT) [9,10]. However, the UK National Institute for Health and Care Excellence (NICE) emphasizes that pharmacotherapy (delivered via DWLSs or in-person services) must be integrated with comprehensive lifestyle interventions and overseen by an MDT to ensure safe and effective patient-centered care [12]. The Juniper UK DWLS is designed to align with these standards by providing MDT guidance and app-based behavioral support, but collecting real-world data from comprehensive DWLS platforms is essential to evaluate the operational effectiveness of such models. Furthermore, the cost of commercial DWLS models, such as the one studied here, often presents a significant financial barrier, potentially leading to the exclusion of patients from lower socioeconomic backgrounds, a known challenge in scaling access to chronic care services [13].”
Methods
Comment 7:
The following area needs improvement in the method section.
- Add references: Please add suitable references for Juniper Central Data Repository in Metabase and the STROBE guidelines.
Response 7: Thank you for noticing this oversight. We have now added both citations to the manuscript.
Comment 8:
- Data Collection Tools: While the manuscript mentions the use of the Juniper app and Metabase for data collection, it could provide more detail on how these tools were validated and whether measures were taken to ensure data accuracy (e.g., verifying patient-reported weight entries).
Response 8: Thank you for this excellent comment. We have now added the following sentence to the third paragraph of the program overview section:
“To enhance data accuracy, the Juniper app employs input validation, flagging implausible weight entries for manual review by the MDT. Additionally, prescribing clinicians mandate weight submission at the 5-month consultation, providing a clinical verification point for patient-reported data.”
Comment 9:
- Medication Titration Schedule: The titration schedule is described, but the rationale for deviations from the standard schedule (e.g., postponing up-titration due to adverse events) could be further clarified to explain how these decisions were made. ​
Response 9: Thank you for this important recommendation. We have now added the following six lines to the end of the medication titration schedule section:
“This clinical judgment involves a risk-benefit assessment: While the standard schedule aims for timely therapeutic effect, prescribers prioritize patient safety and long-term program adherence. A deviation, such as postponing up-titration due to adverse events or missed doses, serves to mitigate the risk of severe or persistent side effects that could otherwise lead to treatment intolerance and discontinuation, thereby undermining therapeutic success.”
Comment 10:
- Handling Missing Data: The use of the Last Observation Carried Forward (LOCF) method for imputing missing weight data is mentioned, but the limitations of this approach (e.g., potential bias) could be discussed more thoroughly in the methods section rather than in the discussion. ​
Response 10: Thank you for this insightful recommendation. We have now added the following 2 sentences to the second paragraph of the Endpoints section:
“It is acknowledged that the LOCF method introduces a known conservative bias, likely underestimating the true mean weight loss for patients who discontinue early. Specifically, this approach fails to capture the common physiological phenomenon of weight rebound that often occurs after the cessation of medicated weight-loss therapy, thereby leading to a more favorable (but potentially inaccurate) mean outcome in the full cohort.”
Comment 11:
Behavioral Metrics: The calculation of behavioral engagement metrics (e.g., weekly weight tracking percentage and health coach message frequency) is described, but the exact process for extracting and analyzing these data from the app is not detailed. ​ This could be clarified further.
Response 11:
Thank you for requesting this clarification. We now added a sentence to the Statistical Analysis section to detail the data extraction and analysis processes:
“All data were extracted from the Juniper central data repository on Metabase using SQL. Data were then uploaded to RStudio, version 2023.06.1+524 (RStudio: Integrated Development Environment for R, Boston, MA, USA) for all analyses and visualisations.”
Comment 12:
- Statistical Analysis: While the statistical methods are described, the manuscript could provide more detail on how the models were validated, including whether any sensitivity analyses were conducted to assess the robustness of the findings.
Response 12:
Thank you for this suggestion. We have added a sentence to the Statistical Analysis section (Section 2.6) to clarify the model validation process and address the query regarding sensitivity analyses. The revised text for section 2.6 now includes the following:
"Model assumptions, including linearity, independence, and homoscedasticity for the linear model and the link function for the logistic model, were assessed and met. No formal sensitivity analyses were conducted, given the retrospective design and the large sample size."
Comment 13:
Results
Table 2 lists "Mild" side effects as the most common level (~57-58%), but the percentages for "Moderate" and "Severe" side effects are missing. ​ This is an omission and should be corrected for completeness.
Response 13:
Thank you for noticing this critical omission. We have now added the moderate and severe side effect data to table 2. We also swapped the mild side effect numbers between the 2 columns as we had them the wrong way around (i.e., adherent cohort in full cohort and vice versa).
Comment 14:
Figure 1: The percentages in the flow chart are inconsistent with the written text.
- The relatively adherent cohort, the percentage is listed as 13.65% of the root, which is unclear. ​ The term "root" is not defined in the document, and it may confuse readers. It would be better to clarify this as "13.65% of the full cohort." ​
- Similarly, for the group that achieved a healthy BMI or target weight, the percentage is listed as 9.68% of the root. ​ This should also be clarified as "9.68% of the full cohort."
Response 14: Thank you for this comment. We have now updated Figure 1 having replaced all mentions of ‘root’ with ‘…of full cohort’
Comment 15:
Figure 2 shows the weight-loss trends for the adherent and full cohorts, but the line labels could be more transparent. ​
- The labels "Adherent patients" and "Full cohort" are correct. Still, it would be helpful to specify that the full cohort data are based on the LOCF (Last Observation Carried Forward) method, as this is mentioned in the text but not in the figure.
Response 15:
Thank you for this important suggestion. We have now added ‘(LOCF)’ to the ‘Full cohort’ label inside the figure.
Comment 16:
Figure 3 shows the churn odds ratio for different tracking categories, but the baseline group (1 weight entry) is labeled "1x" with a p-value of 1, which is incorrect. ​ A p-value of 1 is not meaningful in statistical analysis. This may be a typographical error and should be corrected.
Response 16:
Thank you for noticing this. We have replaced the p-value with the word ‘reference’ in the baseline category and improved the visibility of the plot by giving the baseline a different colour and changing the colour of other bars from orange to blue.
Comment 17:
Discussion
The discussion is well written and aligns with the study's findings and objectives, but it could benefit from more actionable recommendations as follows:
- Specific strategies to improve adherence include: designing interventions to address "burnout" caused by hyper-engagement in the first month, exploring ways to manage patient expectations about weight-loss velocity, and offering financial support or tiered pricing to address socioeconomic barriers for non-Caucasian patients.
Response 17: Thank you for this great recommendation. We have now added the following sentence to the end of the fifth paragraph of the discussion to suggest a strategy to address burnout:
“Specific interventions could include setting realistic, long-term weight-loss goals, emphasizing consistent moderate weekly tracking over intensive daily tracking, and developing automated alerts to flag patients who exhibit high initial weight-loss velocity, allowing for timely, supportive coach intervention to manage expectations and mitigate burnout.”
We also added this sentence (in bold) to the penultimate paragraph of the discussion to suggest a strategy for overcoming socioeconomic barriers:
“Finally, the discovery that a disproportionate number of non-Caucasian patients failed to adhere to the program for 12 months is consistent with previous research on DWLSs and chronic care services in general [27, 28]. Although income-related data were not available, It is likely that this trend is a reflection of Juniper UK’s high monthly fee. Addressing these disparities may require structural interventions, such as exploring financial support or tiered pricing models for underserved populations.”
Comment 18:
- More context on how DWLS clinicians can address this issue, such as educating patients about realistic weight-loss trajectories, implementing pacing strategies to encourage moderate engagement and sustainable habits, and developing tools to identify and support patients at risk of burnout early in the program.
Response 18: Thank you for this quality suggestion. We have now added the following sentence to the end of the fifth paragraph of the discussion to suggest a strategy to address burnout:
“Specific interventions could include setting realistic, long-term weight-loss goals, emphasizing consistent moderate weekly tracking over intensive daily tracking, and developing automated alerts to flag patients who exhibit high initial weight-loss velocity, allowing for timely, supportive coach intervention to manage expectations and mitigate burnout.”
Comment 19:
- Potential interventions to address these disparities include offering culturally tailored support and resources, investigating socioeconomic factors more thoroughly in future studies, and exploring partnerships with public health organizations to subsidize costs for underserved populations.
Response 19: Thank you for this important comment. We have now added the following sentence (bold) to the final paragraph of the discussion:
“The sample was also overrepresented by female and Caucasian patients, and given the program fee, would have likely excluded patients of lower socioeconomic status. These factors limit the study’s generalizability. Future studies should investigate the role of culturally tailored support and financial barriers in DWLS adherence among diverse populations.”
We have also added this sentence to the penultimate paragraph of the discussion:
“Although income-related data were not available, It is likely that this trend is a reflection of Juniper UK’s high monthly fee. Addressing these disparities may require structural interventions, such as exploring financial support or tiered pricing models for underserved populations.”
Comment 20:
- Propose specific improvements for future studies, such as conducting prospective studies to establish causation and using alternative methods to handle missing data, such as multiple imputation. Recruiting a more diverse and representative sample to improve generalizability. Collecting more complete data on discontinuation reasons and socioeconomic factors
Response 20:
Thank you for this insightful recommendation. We have now added the following sentence to the end of the conclusion:
Future prospective studies are needed to establish causation and test targeted adherence interventions. Methodologically, these studies should employ alternative imputation techniques, such as multiple imputation, to more accurately handle missing data. Furthermore, future work must prioritize recruiting a more diverse and representative sample and should incorporate robust mechanisms to prospectively capture patient discontinuation reasons and detailed socioeconomic data.
Comment 21:
- Evidence-based strategies for DWLS providers include setting realistic weight-loss goals for patients, monitoring early engagement, and intervening when hyper-engagement is detected. ​Enhancing health coach training to address performance anxiety and burnout, and developing targeted interventions for patients with high initial BMI or multiple comorbidities.
Response 21:
Thank you for this important comment. We have integrated these points into both the discussion and conclusion.
Specifically, the discussion now includes a detailed explanation of how clinicians can manage hyper-engagement and prevent burnout:
"Specific interventions could include setting realistic, long-term weight-loss goals, emphasizing consistent moderate weekly tracking over intensive daily tracking, and developing automated alerts to flag patients who exhibit high initial weight-loss velocity, allowing for timely, supportive coach intervention to manage expectations and mitigate burnout."
And the conclusion was strengthened to emphasize that the predictive models should inform, rather than dictate, practice, and to address the need for targeted, equity-focused interventions:
"Future prospective studies are needed to establish causation and test targeted adherence interventions... Furthermore, future work must prioritize recruiting a more diverse and representative sample and should incorporate robust mechanisms to prospectively capture patient discontinuation reasons and detailed socioeconomic data."
Reviewer 2 Report
Comments and Suggestions for Authors
Title:
Preferably to signal the context and methodological design in the title
Suggestion: “12-Month Weight Loss and Adherence in a UK Tirzepatide-Supported Digital Obesity Service: A Retrospective Cohort Analysis”
Abstract:
Please clarify what distinguishes this digital service from standard obesity care to contextualize the study’s relevance. Also, clarify the meaning of “digital weight-loss service”? Is it app-based, clinician-supervised?
Please ensure the research gap is stated more clearly
Please specify the main covariate groups included in the models
Please clarify whether differences between the adherent and full cohorts are statistically significant
Please clarify what “sustainable, moderate behavioral pacing” requires operationally
Introduction:
Please expand on why mechanistic superiority matters in a real-world digital service context
Please clarify whether Juniper's model aligns fully with NICE standards
Please clarify whether effectiveness refers to weight outcomes only or includes engagement, safety, or clinical parameters
Please specify whether the study design allows causal inference or only association
Materials and Methods:
Please justify the unusual study end date (January 1 to November 11, 2024) not the full year?
Please explain how medication adherence (actual injections) was monitored beyond order counts
Please explain whether dose-related pricing created socioeconomic differences between adherent and non-adherent groups
Results:
Please align the time window in results (351–379 days) with the definition in methods (355–375 days
Please describe whether baseline characteristics differed significantly between adherent and non-adherent groups
Please add a column in Table 1 with p-values or standardized mean differences comparing full vs adherent cohorts to quantify selection differences
Please label in Figure 1 how many patients were included in each analysis
Please explain why intermediate high doses (10 and 12.5 mg) did not differ from 15 mg, and discuss whether sample size for these strata was limited
Please elaborate on why prior use of modern weight-loss medications predicts lower adherence
Discussion:
Please describe real-world outcomes as “effectiveness” not “efficacy” given the observational design
You can strengthen the contextualisation of your primary weight-loss findings with a brief Tirzepatide narrative synthesis after discussing the similarity between your 12-month outcomes and RCT evidence. Suggestion:
“Furthermore, the finding that 98.85 percent of adherent patients achieved the clinically significant 5 percent weight-loss threshold suggests that medicated DWLSs deliver reliable outcomes to adherent patients. This aligns with published narrative syntheses describing Tirzepatide’s dual GIP/GLP-1 receptor mechanism as a key driver of its robust weight-loss effects across diverse clinical trials and obesity-management settings (https://doi.org/10.3390/obesities5020026).
Please state that the proposed mechanisms (burnout, performance anxiety, social-media-driven expectations) are hypothesized explanations, not empirically tested in this dataset
Please suggest how future program iterations could improve the capture of discontinuation reasons
Please discuss additional structural factors that may contribute to ethnic disparities in DWLS adherence.
Please suggest how future work could use more robust longitudinal methods
Please expand the limitation regarding observational design to include the risk of unmeasured confounding
Please indicate how future studies could prospectively capture discontinuation reasons
Conclusions:
Please replace “confirms”, which implies causality with “suggests”
Please use “clear clinical guidance”, as prediction models in observational studies should inform hypothesis generation rather than dictate practice. Suggestion: “The study’s predictive models offer preliminary insights that can help inform clinical decision-making.”
Please add a brief statement acknowledging the need for equity-focused interventions
Please suggest evaluating both digital engagement strategies and clinical management approaches to address multifactorial drivers of dropout
General:
- Please make sure English proofreading is completed to maintain grammatical consistency, avoid spelling mistakes, tense consistency, and ensure a uniform, concise academic tone throughout the text.
- Use consistent terminology throughout the manuscript.
- Please ensure that the in-text citation and reference list follow journal formatting guidelines
Comments on the Quality of English LanguagePlease ensure English proofreading is completed to maintain grammatical consistency, avoid spelling mistakes, tense consistency, and ensure a uniform, concise academic tone throughout the text
Author Response
Comment 1:
Preferably to signal the context and methodological design in the title Suggestion: “12-Month Weight Loss and Adherence in a UK Tirzepatide-Supported Digital Obesity Service: A Retrospective Cohort Analysis”
Response 1: Thank you for this excellent recommendation. We have now incorporated elements of your suggestion with that of another reviewer to develop the following title: 12-Month Weight Loss and Adherence Predictors in a real-world UK Tirzepatide-Supported Digital Obesity Service: A Retrospective Cohort Analysis
Comment 2:
Abstract: Please clarify what distinguishes this digital service from standard obesity care to contextualize the study’s relevance. Also, clarify the meaning of “digital weight-loss service”? Is it app-based, clinician-supervised? Please ensure the research gap is stated more clearly
Response 2:
Thank you for this important suggestion. We have now added 2 sentences (in bold) to the background section of the abstract to address this:
“Background: Obesity management is evolving with the integration of dual GIP/GLP-1 receptor agonists (Tirzepatide) into comprehensive Digital Weight-Loss Services (DWLSs). This model leverages virtual, app-based multidisciplinary care (MDT) to deliver continuous, supervised treatment, distinguishing it from traditional, intermittent clinic-based care. While clinical trials demonstrate high efficacy, real-world data are necessary to evaluate long-term adherence and identify predictive markers for patient persistence in these scalable care models. Specifically, there is a knowledge gap regarding the specific behavioral factors that govern 12-month persistence in these comprehensive, medicated DWLS settings. This study retrospectively assessed the 12-month effectiveness and adherence of a Tirzepatide-supported DWLS and identified demographic, clinical, and behavioral predictors of weight loss and program attrition.
Comment 3:
Please specify the main covariate groups included in the models:
Response 3: Thank you for this comment. We have now added the following clause to the methods section of the abstract:
“Binary logistic and multiple linear regression models identified predictors of adherence and weight loss, respectively, using a comprehensive set of demographic, clinical (e.g., BMI, comorbidities), and behavioral variables.”
Comment 4:
Please clarify whether differences between the adherent and full cohorts are statistically significant
Response 4: Thank you for this comment. We have now added this sentence to the results section of the abstract:
“This difference in 12-month mean weight loss was statistically significant (p < 0.001).”
Comment 5:
Please clarify what “sustainable, moderate behavioral pacing” requires operationally
Response 5:
Thank you for this comment. We have now added the following 2 clauses (in bold) to the conclusion of the abstract:
“However, patient persistence remains the primary translational challenge. As consistent weekly engagement (tracking, coaching) is the strongest predictor of success, clinical strategies should prioritize promoting sustainable, moderate behavioral pacing (i.e., emphasizing consistent weekly engagement over intensive daily tracking and rapid early weight loss) to mitigate attrition risk and optimize the public health effectiveness of medicated DWLSs.”
Comment 6:
Introduction: Please expand on why mechanistic superiority matters in a real-world digital service context
Response 6: Thank you for this suggestion. We have now added the following sentence to the end of the second paragraph of the introduction:
“The high intrinsic efficacy of this class is particularly relevant in a scalable DWLS model, as the platform's primary goal is to ensure that the medication’s maximal therapeutic potential is reliably translated into real-world outcomes across a large patient population.”
Comment 7:
Please clarify whether Juniper's model aligns fully with NICE standards
Response 7: Thank you for this important request. We have edited the section on NICE standards in the third paragraph of the introduction. This now reads as follows:
“However, the UK National Institute for Health and Care Excellence (NICE) emphasizes that pharmacotherapy (delivered via DWLSs or in-person services) must be integrated with comprehensive lifestyle interventions and overseen by an MDT to ensure safe and effective patient-centered care [12]. The Juniper UK DWLS is designed to align with these standards by providing MDT guidance and app-based behavioral support, but collecting real-world data from comprehensive DWLS platforms is essential to evaluate the operational effectiveness of such models.”
Comment 8:
Please clarify whether effectiveness refers to weight outcomes only or includes engagement, safety, or clinical parameters
Response 8:
Thank you for requesting this clarification. We have edited the first sentence of the final paragraph of the intro to explicitly state the study’s focus is on weight loss:
“This study addresses the existing knowledge gap by retrospectively analyzing a large cohort of patients enrolled in the Juniper UK DWLS to determine the clinical effectiveness (weight-loss outcomes) of the comprehensive Tirzepatide-supported program over 12 months.
Comment 9:
Please specify whether the study design allows causal inference or only association
Response 9:
Thank you for this important request. We have now added the following sentence to the final paragraph of the introduction:
“As a retrospective cohort study, this design allows for the identification of associations and predictive factors, but not causal inference.”
Comment 10:
Materials and Methods: Please justify the unusual study end date (January 1 to November 11, 2024) not the full year?
Response 10: Thank you for this important question. We have now added the following sentence to the study design section to justify the end date:
“The latter date was selected on the basis of it falling exactly 12 months after the data extraction date (11 November 2025).”
Comment 11:
Please explain how medication adherence (actual injections) was monitored beyond order counts
Response 11: Thank you for this sharp observation. We have now added a sentence to the end of the endpoints paragraph to clarify this:
“It is important to note that, given the asynchronous nature of the DWLS, medication adherence is measured via the count of pharmacy orders, which serves as a proxy for medication consumption. Actual self-administration (injection) is not directly monitored.”
Along with the following clause to the limitations paragraph in the discussion:
“and medication adherence was measured via the count of pharmacy orders, which was only a proxy for medication consumption and therefore may not have been completely accurate.”
Comment 12:
Please explain whether dose-related pricing created socioeconomic differences between adherent and non-adherent groups
Response 12: Thank you for this important comment. We have now added the following four lines to the program cost sub-section:
“Due to the lack of income data in this retrospective analysis, we were unable to assess whether the higher cost associated with higher Tirzepatide doses acted as a financial barrier contributing to socioeconomic differences between the adherent and non-adherent cohorts.”
Comment 13:
Results: Please align the time window in results (351–379 days) with the definition in methods (355–375 days
Response 13:
Thank you for noticing this inconsistency. We have now corrected the 351-379 day error.
Comment 14:
Please describe whether baseline characteristics differed significantly between adherent and non-adherent groups
Response 14: Thank you for this excellent recommendation. We have now added the following two sentences to the end of the first paragraph of the results section:
“Statistically significant baseline differences were observed between the adherent and full cohorts. The adherent cohort was older, had a higher proportion of male patients, and a higher proportion of Caucasian patients, suggesting a selection effect for long-term persistence in the program.”
Comment 15:
Please add a column in Table 1 with p-values or standardized mean differences comparing full vs adherent cohorts to quantify selection differences.
Response 15:
Thank you for this suggestion. We have now also added a p-value column to table 1. We also added the following sentence to the statistical analysis section to explain how we calculated these p-values:
“Continuous variables (age, initial BMI, initial weight) were compared using the independent-samples Welch's t-test and categorical variables (sex at birth, ethnicity, and comorbidities) were compared using the Chi-squared (x2) test to determine if baseline distributions differed significantly between the adherent and full cohorts.”
Comment 16:
Please label in Figure 1 how many patients were included in each analysis
Response 16:
Thank you for this comment. We have now added text to the first box to stress that this is the ‘Full cohort’. The lowest box is titled ‘Adherent 12-month patients’
Comment 17:
Please explain why intermediate high doses (10 and 12.5 mg) did not differ from 15 mg, and discuss whether sample size for these strata was limited
Response 17:
Thank you for this insightful comment. As this is an interpretation of the results, we have now added a possible explanation to the penultimate discussion paragraph:
“Regarding the medication titration, the finding that intermediate doses (10mg and 12.5mg) did not significantly differ from the highest dose (15mg) in predicting 12-month weight loss warrants further investigation. This suggests that the maximum weight loss response may plateau at or before the 10mg dose, or alternatively, that the statistical power within these specific dose strata was insufficient to detect a smaller difference when compared to the reference group.”
Comment 18:
Please elaborate on why prior use of modern weight-loss medications predicts lower adherence
Response 18:
Thank you for this important comment. As this is an interpretation of the results, we have now added a possible explanation to the penultimate discussion paragraph:
“The negative association between previous use of modern weight-loss medications (GLP-1 RA or dual agonist) and 12-month adherence (beta = -0.35, p < 0.001) is notable. This correlation may indicate that this sub-population, being less treatment-naïve, may harbor higher, potentially unrealistic expectations about the speed or extent of weight loss, or they may represent patients who are prone to cycling through various pharmacological treatments without establishing long-term behavioral persistence.”
Comment 19:
Discussion: Please describe real-world outcomes as “effectiveness” not “efficacy” given the observational design
Response 19:
Thank you for noticing this error. We have now replaced the references to ‘efficacy’ in the abstract and discussion with “effectiveness”.
Comment 20:
You can strengthen the contextualisation of your primary weight-loss findings with a brief Tirzepatide narrative synthesis after discussing the similarity between your 12-month outcomes and RCT evidence. Suggestion: “Furthermore, the finding that 98.85 percent of adherent patients achieved the clinically significant 5 percent weight-loss threshold suggests that medicated DWLSs deliver reliable outcomes to adherent patients. This aligns with published narrative syntheses describing Tirzepatide’s dual GIP/GLP-1 receptor mechanism as a key driver of its robust weight-loss effects across diverse clinical trials and obesity-management settings (https://doi.org/10.3390/obesities5020026).
Response 20:
Thank you for this excellent recommendation. We have now added the following 5 lines to the 2nd paragraph of the discussion:
“Furthermore, the finding that 98.85 percent of adherent patients achieved the clinically significant 5 percent weight-loss threshold suggests that medicated DWLSs deliver reliable outcomes to adherent patients. This aligns with published narrative syntheses describing Tirzepatide’s dual GIP/GLP-1 receptor mechanism as a key driver of its robust weight-loss effects across diverse clinical trials and obesity-management settings [18].”
Comment 21:
Please state that the proposed mechanisms (burnout, performance anxiety, social-media-driven expectations) are hypothesized explanations, not empirically tested in this dataset
Response 21:
Thank you for this important comment. We have now added the following sentence to the 5th paragraph of the discussion to address this point:
“It must be noted that these proposed psychological mechanisms (burnout, performance anxiety) are hypothesized based on the observed statistical associations and prior literature and were not empirically tested or directly measured in this retrospective dataset.”
Comment 22:
Please suggest how future program iterations could improve the capture of discontinuation reasons
Response 22:
Thank you for this suggestion. We have now added the following sentence to the sixth paragraph of the discussion section:
“Future program iterations should explore incentivizing the completion of exit surveys (e.g., small rewards) or utilizing brief, structured, automated messaging sequences (e.g., SMS/email) triggered immediately upon app cessation, which may yield higher response rates for capturing discontinuation reasons.”
Comment 23:
Please discuss additional structural factors that may contribute to ethnic disparities in DWLS adherence.
Response 23: Thank you for this insightful comment. We have now added the following sentence to the penultimate paragraph of the discussion:
“Beyond cost, potential structural factors contributing to lower adherence may include differences in digital health literacy, access to high-speed internet, or a lack of culturally tailored MDT support and resources [26]. Addressing these disparities may require structural interventions, such as exploring financial support or tiered pricing models for underserved populations.”
Comment 24:
Please suggest how future work could use more robust longitudinal methods
Response 24:
Thank you for this comment. We have now added the following sentence to the final paragraph of the discussion:
“Future prospective studies should employ more robust longitudinal techniques, such as multiple imputation, to more accurately handle missing data.”
Comment 25:
Please expand the limitation regarding observational design to include the risk of unmeasured confounding
Response 25:
Thank you for this recommendation. We have now edited the first sentence of the final discussion paragraph to read as follows:
“This study’s interpretation is limited by its retrospective design, which can only establish correlations, not causation and is subject to the inherent risk of unmeasured confounding from unrecorded variables (e.g., patient-reported dietary data, true socioeconomic status) that may influence both adherence and weight loss.”
Comment 26:
Please indicate how future studies could prospectively capture discontinuation reasons
Response 26: Thank you for this comment. We have now added the following sentence to the 6th paragraph of the discussion:
“Alternatively, prospective studies could be run, whereby investigators call every patient who stops the program early to record their primary discontinuation reason.”
Comment 27:
Conclusions: Please replace “confirms”, which implies causality with “suggests”
Response 27:
Thank you for noticing this error. We have now applied this change.
Comment 28:
Please use “clear clinical guidance”, as prediction models in observational studies should inform hypothesis generation rather than dictate practice.
Response 28:
Thank you for this important comment. The sentence now reads as follows:
“The study’s predictive models offer preliminary insights that can help inform clinical decision-making for overcoming this challenge.”
Comment 29:
Suggestion: “The study’s predictive models offer preliminary insights that can help inform clinical decision-making.” Please add a brief statement acknowledging the need for equity-focused interventions Please suggest evaluating both digital engagement strategies and clinical management approaches to address multifactorial drivers of dropout
Response 29:
Thank you for this excellent suggestion. We have now added the following sentence to the 2nd paragraph of the conclusion section:
“Furthermore, the findings underscore the need for future program development to include equity-focused interventions to address socioeconomic and demographic barriers to long-term persistence.”
Round 2
Reviewer 1 Report
Comments and Suggestions for Authors
I appreciate the authors for the extensive revision, which has a greater impact on the manuscript for readers. However, the following comment has not yet been addressed by the authors. I'm looking forward to your reply for the same.
Comment 4:
- Comparison with Other DWLS: The background could benefit from a more detailed comparison of the Juniper UK DWLS with other DWLS models (if any), particularly regarding their structures, costs, and integration of pharmacotherapy and lifestyle interventions.
Author Response
Comment:
I appreciate the authors for the extensive revision, which has a greater impact on the manuscript for readers. However, the following comment has not yet been addressed by the authors. I'm looking forward to your reply for the same.
- Comparison with Other DWLS: The background could benefit from a more detailed comparison of the Juniper UK DWLS with other DWLS models (if any), particularly regarding their structures, costs, and integration of pharmacotherapy and lifestyle interventions.
Response:
We thank the reviewer for this insightful suggestion. We agree that providing a taxonomy of the current Digital Weight-Loss Service (DWLS) landscape helps contextualize the Juniper UK model and its emphasis on multidisciplinary care. Accordingly, we have revised the Introduction to include a detailed comparison of three primary DWLS frameworks:
- Lifestyle-only models
- Medication-first telehealth platforms
- Comprehensive MDT-integrated models
The following 11 lines have been added to the end of paragraph 3/start of paragraph 4 (in bold) along with 6 references to the reference list.
The introduction of these highly effective pharmaceuticals has coincided with the rapid expansion of digital health solutions. Medicated Digital Weight Loss Services (DWLSs) offer a critical pathway to scale access to obesity care. These platforms break down common geographic and temporal barriers to continuous medical management [9, 10], and virtual care environments have been noted to potentially foster greater comfort and openness in patient-provider communication regarding sensitive health topics [11]. Unique features typically include app-based progress tracking, personalized lifestyle support, and asynchronous, on-demand communication with a multidisciplinary care team (MDT) [9,10]. These latter features, with the possible exception of MDT support, predate medicated DWLSs. Programs such as Noom and Oviva are long-standing DWLS providers that have utilized standalone lifestyle interventions to drive weight loss, i.e., without pharmacotherapy [12, 13]. These services remain widely utilized and have demonstrated effectiveness in achieving modest weight loss through habit formation.
With the clinical validation of GLP-1 RAs for weight loss, the landscape has bifurcated. One model involves 'medication-first' platforms, which function as streamlined telehealth services focusing on prescription and safety follow-ups [14, 15]. In contrast, 'comprehensive' models such as Ro attempt to replicate the MDT environment of clinical trials by mandating a combination of medication and intensive lifestyle intervention [16]. The Juniper UK model follows this comprehensive structure. While the operational costs of commercial DWLS models are rarely detailed, it is well established that medicated programs incur significantly higher monthly fees than lifestyle-only services [17].